# TLM: Token-Level Masking for Transformers

**Yangjun Wu,**[*] **Kebin Fang**[*]**, Dongxiang Zhang,**[†] **Han Wang, Hao Zhang, Gang Chen**

Zhejiang University

{yjwu, fkb}@zjuici.com,

{22021066, zhangdongxiang, cg }@zju.edu.cn,

hz283@sussex.ac.uk

## Abstract

Structured dropout approaches, such as attention dropout and DropHead, have been investigated to regularize the multi-head attention mechanism in Transformers. In this paper, we propose a new regularization scheme based on token-level rather than structure-level to reduce overfitting. Specifically, we devise a novel **T**oken-**L**evel **M**asking (TLM) training strategy for Transformers to regularize the connections of self-attention, which consists of two masking techniques that are effective and easy to implement. The underlying idea is to manipulate the connections between tokens in the multi-head attention via masking, where the networks are forced to exploit partial neighbors' information to produce a meaningful representation. The generality and effectiveness of TLM are thoroughly evaluated via extensive experiments on 4 diversified NLP tasks across 18 datasets, including natural language understanding benchmark GLUE, ChineseGLUE, Chinese Grammatical Error Correction, and data-to-text generation. The results indicate that TLM can consistently outperform attention dropout and DropHead, e.g., it increases by 0.5 points relative to DropHead with BERT-large on GLUE. Moreover, TLM can establish a new record on the data-to-text benchmark Rotowire (18.93 BLEU). Our code will be publicly available at https://github.com/Young1993/tlm.

## 1 Introduction

In recent years, a variety of pre-trained language models based on the Transformer (Vaswani et al., 2017) architecture have been presented, such as BERT, GPT (Brown et al., 2020), and T5 (Raffel et al., 2022). These models push state-of-the-art forward in numerous NLP tasks.

With the rapid growth of model parameters, deep neural networks are highly likely to encounter overfitting challenges because supervised data is usu-

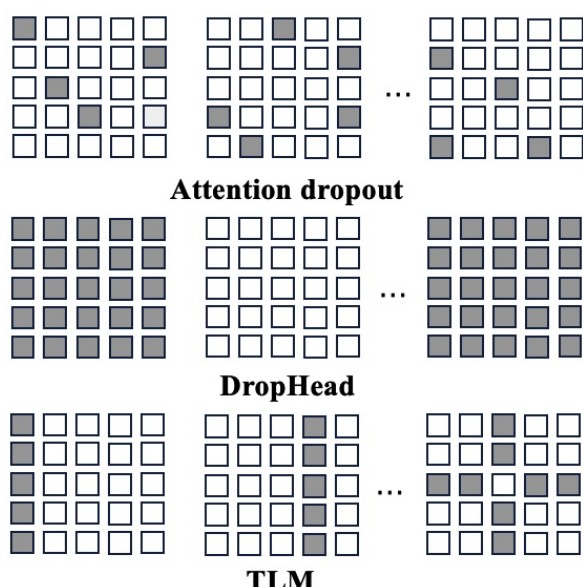

Figure 1: Illustrations of the attention score with Attention dropout, DropHead, and TLM. The row denotes *Query*, and the column represents *Key*. Attention dropout randomly drops some attention weights. DropHead directly drops entire attention heads. Regarding TLM, Self-masking (left) denotes that the scores for the masked column are invalid. Siblings-masking (right) means the scores for the row and column are useless except for the masked token itself.

ally expensive and insufficient for large language models. This problem could cause the degradation of model generalization. To address this issue, regularization methods, such as dropout (Srivastava et al., 2014) and subsequent research (Wan et al., 2013; Fan et al., 2020; liang et al., 2021) have been developed from a structural perspective, which trained with "thinned" subnetworks. The feature of dropout is that it randomly drops units from the neural networks during training, preventing units from co-adapting too much.

To further mitigate overfitting for Transformers, structured methods such as DropHead (Zhou et al., 2020) are proposed to drop entire attention heads in the attention mechanism with the purpose of pre-

---

[*] Equal contribution.

[†] Corresponding author.

venting a small subset of heads from dominating the whole model. Dropping entire attention heads may result in losing a significant amount of feature information. Attention dropout is the application of the dropout technique to the attention mechanism. It arbitrarily drops some attention weights in the matrix calculation of self-attention. However, experiments in DropHead and our preliminary trials (shown in Table 1) demonstrate that the difference is not obvious with or without attention dropout.

In this work, we introduce a novel regularization scheme based on token-level instead of a structural perspective. This method, **T**oken-**L**evel **M**asking (TLM), is a training technique to regularize the connections among tokens during the attention calculation in each layer of Transformer blocks. Specifically, considering the example shown in Fig.1 and 2, TLM contains two techniques: 1) Siblings-masking. The first step of this method is that we use random function[1] to select a percentage of the tokens in the *k-th* layer, e.g., the masked token 'I' (the gray block in Fig.2) is excluded from the calculation of attention weights among the sibling tokens but copies itself, and its neighboring tokens considers other siblings in addition to 'I'. Then, we feed the Feed-Forward Network with the attention output to obtain the new hidden state as the input of the next layer. 2) For Self-masking, we borrow the idea from CBOW (Mikolov et al., 2013) where its attention score is entirely contributed by others. The difference with Siblings-masking is the masked token 'I' is forbidden to attend to attention computation. In the training phase, we randomly invoke one of two masking strategies at each batch with a 50-50 chance[2]. In this manner, the networks are forced to utilize partial neighbors' attention information, not the whole (i.e. the connections between the masked tokens and their neighboring tokens are invalid, which are implemented by assigning a large negative number in the matrix of attention weights). This scheme introduces a bottleneck that the nets should work hard to become robust and produce a meaningful representation.

To confirm the effectiveness of our approach, we conducted extensive experiments on 18 popular datasets. The tasks range from English natural language understanding benchmark GLUE,

| Model | QQP | RTE |
|---|---|---|
| BERT -w Attention-dropout | 87.0 | 61.8 |
| BERT -w/o Attention-dropout | 86.9 | 62.0 |

Table 1: Results of BERT w and w/o attention-dropout on QQP and RTE. The descriptions of datasets are available in section 4.1.

ChineseGLUE, and Chinese Grammatical Error Correction, to data-to-text generation. The experimental results demonstrate that our TLM with the backbones can substantially improve performance. Particularly, our method with BERT-base/BERT-large boosts the score of DropHead from 79.2 to 79.9 and 81.7 to 82.2 on GLUE, and it achieves a new state-of-the-art performance (18.93 BLEU) on data-to-text generation. Further experimental analyses demonstrate that our TLM is more effective in alleviating overfitting than the baselines.

Our main contributions are summarized as follows:

- To reduce overfitting, we present TLM, a novel, simple yet effective training technique to refine the self-attention computation flow in the multi-head attention mechanism without modifying the structure of Transformer models.

- TLM can seamlessly integrate with pre-trained Transformer models without extra cost. The experiments on 18 popular datasets indicate that TLM can lead to consistency improvements compared to the strong baselines.

- Further analyses demonstrate that TLM can reduce overfitting and enhance the robustness of the networks.

## 2 Related Work

**Mask language modeling.** In BERT(Devlin et al., 2019), 15% of input tokens are selected, and 80% of those selected tokens are replaced with the special token [MASK], while 10% remain unchanged and the remaining 10% are randomly replaced. However, this random masking and replacement are only done once during data pre-processing, resulting in a mismatch between pre-training and fine-tuning. RoBERTa (Liu et al., 2019) duplicates training data 10 times to address this issue, but this requires more training steps. In

---

[1]Bernoulli function: `https://pytorch.org/docs/stable/generated/torch.bernoulli.html?highlight=bernoulli`

[2]For simplicity, we conduct most of the experiments with 50-50 chance and ablate the proportion in Appendix D.

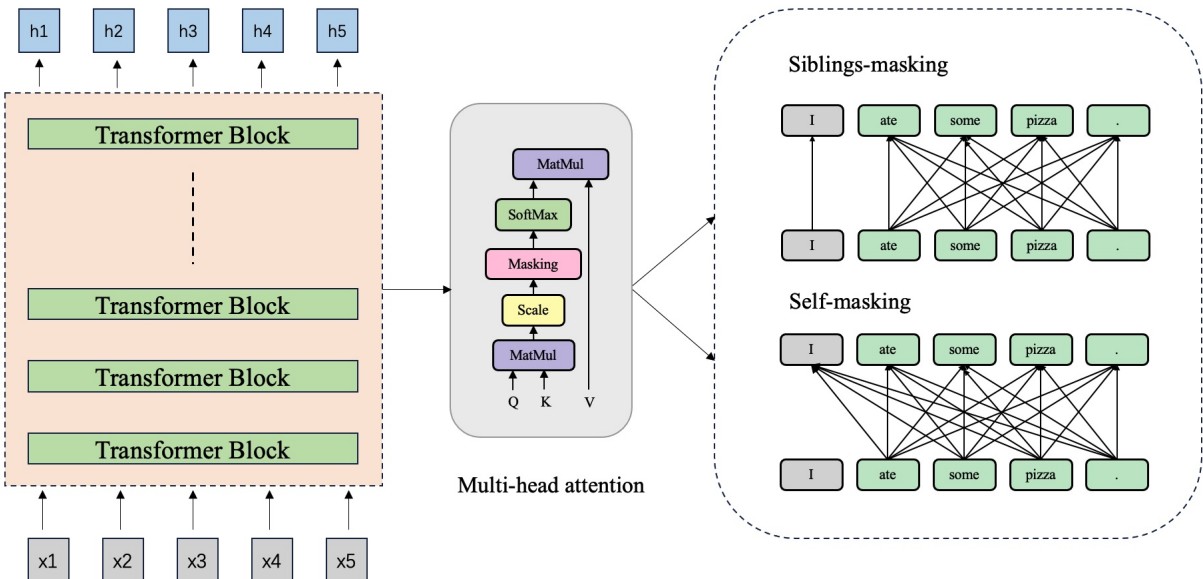

Figure 2: The computing flow of TLM for the sentence '*I ate some pizza.*'. Our TLM can be employed on the encoder, encoder-decoder, and decoder architecture in the Transformer.

contrast, our proposed method modifies the attention computation flow in each layer without requiring additional data or a special token.

**Attention Mask.** Attention Mask in Transformers, which is originally used to ignore those invaluable tokens, called padding tokens, in order to ensure that all the sequences are the equal length for each batch; or to prevent the networks from seeing the tokens' subsequent position in the auto-regressive decoder, such as UniLM (Dong et al., 2019), OPT(Zhang et al., 2022), and LLaMA(Touvron et al., 2023). In this work, we also employ the attention mask with the purpose of mitigating overfitting, i.e., we remove some attention links among tokens during training and keep the same as the standard Transformer in inference.

**Regularization.** Common regularization methods include L1 and L2 regularization, dropout, early stopping, and data augmentation. Dropout (Srivastava et al., 2014) is a very popular training technique that aims to avoid overfitting by arbitrarily dropping some units. LayerDrop (Fan et al., 2020) randomly drops some entire substructures or components. Data dropout (Iyyer et al., 2015) is performed on the input data level as data augmentation. Compared to the previous methods, our approach aims to carefully control the connections between tokens in the multi-head attention mechanism.

## 3 Approach

In this section, we first review the self-attention computing workflow of the Transformer and then describe our TLM in more detail.

### 3.1 Multi-head Attention in Transformer

In the research of the Transformer (Vaswani et al., 2017), the calculation for the vanilla attention is formulated as follows:

$$Attn(Q, K, V) = softmax(\frac{S(QK)}{\sqrt{d_{emb}/H}})V \quad (1)$$

$$S(QK) = MatMul(QK^T) \quad (2)$$

where the queries $Q$, keys $K$, and values $V$ are all matrices, where the shapes are $Batch\ size \times sequence\ length \times d_{emb}$. $d_{emb}$, and $H$ denote the dimension of embedding and the number of heads, separately.

There are three types of multi-head attention: 1) **Self-Attention** usually denotes the self-attention in the encoder, which means all of the keys ($K$), values ($V$), and queries ($Q$) come from the same place in a self-attention layer. Thus, every position in the encoder can attend to the attention computing of all positions. 2) **Cross-Attention** is applied in the encoder-decoder layer, and its feature is that queries ($Q$) come from the decoder layer, and the memory keys/values come from the output of the encoder. In this manner, each position in the decoder can be present at the attention computing of

**Algorithm 1** TLM Training Procedure
---
1: Initialize model with parameters $\omega$.
2: **while** not converged **do**
3:    **for** i = 1 to M layers **do**
4:       Obtain attention weights $S(QK)$ (Eq.2)
5:       Select tokens to mask (Eq.3)
6:       Expand $Att\hat{n}\_M$ into matrix $M$ (Eq.4)
7:       Calculate $A\hat{t}tn(Q, K, V)$ (Eq.5)
8:       Obtain the hidden state $h_t$
9:    **end for**
10: **end while**
---

all positions in the input sequence. 3) **Masked-Attention** is the form of auto-regression in the decoder. The self-attention layers allow each position in the decoder to attend to all places in the decoder up to the current position.

### 3.2 TLM

The core computing procedure of TLM is shown in Fig.2 and Algorithm (1). In a nutshell, we modify the computing of self-attention by adding a novel step to control the connections between tokens, which in turn influences the attention scores and contextual semantic representation. We utilize TLM in the training phase while keeping the attention calculation as the vanilla during testing.

In the training phase, we first compute attention weights $S(QK)$ by performing Eq.2, and $S(QK)$ denotes the similarity between the queries $Q$ and keys $K$ for the input tokens. Then, the random function *Bernoulli* is executed to select a fixed rate $R$ of masked tokens in the *k-th* layer of Transformer blocks at each batch.

$$Att\hat{n}\_M = Bernoulli(Attn\_M, R) \quad (3)$$

Where $Attn\_M$ refers to the vector of attention-mask [3]. The tokens selected as masked tokens will be stored in memory with the attention mask value of 0. When the rate $R$ is set to $0.1$, which denotes 10% of tokens would be masked.

To fit the identical dimension as the attention weight $S(QK)$, we expand the attention mask vector $Att\hat{n}\_M$ into the matrix $M$:

$$M = Extend(Att\hat{n}\_M) \quad (4)$$

Here, $M \in \mathbb{R}^{B \times H \times N \times N}$. $B$, $H$, and $N$ refer to batch size, the number of self-attention heads,

---
[3]Attention-mask, is abbreviated as $Attn\_M$, $Attn\_M = [1, 1, ...0]$. The values equal 1 denoting the input tokens, or 0 when it belongs to padding token or masked token.

and the max input sequence length, respectively. The weights of masked tokens in $M$ are set to the minimum value of the tensor. Then, we can modify the Eq.1 and 2 as follows:

$$A\hat{t}tn(Q, K, V) = softmax(\frac{\hat{S}(QK)}{\sqrt{d_{emb}/H}})V \quad (5)$$

$$\hat{S}(QK) = S(QK) + M \quad (6)$$

The attention scores of masked connections among tokens are very large negative numbers by performing Eq.6, so their weights equal 0 after executing *softmax*. This makes the connections to the masked tokens ineffective in influencing the current masked token.

Next, we feed the Feed-Forward Network with $A\hat{t}tn(Q, K, V)$ to obtain the hidden state $h_t$. We recursively invoke the identical operation until all the layers have been traversed and yield the final output tensors.

## 4 Experiments

To verify the generality and effectiveness of our proposed TLM, we perform extensive experiments on a wide variety of tasks with 18 benchmark datasets, including the English natural language understanding benchmark GLUE (10 datasets), ChineseGLUE (6 datasets), Chinese Grammatical Error Correction (1 dataset), and data-to-text generation (1 dataset). Note that our method can be utilized both in the pre-training and fine-tuning, but we only estimate TLM during the fine-tuning in this work due to limited computational resources. In the following, we present the key findings, and more details can be found in the Appendix.

### 4.1 English Language Understanding

**Dataset.** GLUE benchmark is a collection of diverse natural language understanding tasks introduced by (Wang et al., 2018). GLUE consists of three types of tasks: single-sentence, similarity and paraphrase, and inference tasks. Single-sentence tasks require models to predict the grammaticality or sentiment of a given sentence. Similarity and paraphrase tasks involve determining the degree of semantic equivalence between sentence pairs. Inference tasks aim to capture the entailment relationship between sentences.

**Model and Training.** For a fair comparison, we choose BERT as the backbone and train it

| Model | CoLA | SST-2 | MRPC | STS-B | QQP | MNLI-m | MNLI-mm | QNLI | RTE | WNLI | AVG | STD |
|---|---|---|---|---|---|---|---|---|---|---|---|---|
| **BERT-small** | | | | | | | | | | | | |
| w/o Att-dropout | 27.5 | 89.3 | 83.2 | 78.9 | 86.9 | 77.5 | 76.8 | 86.2 | 62.0 | 62.1 | 73.0 | 2.8e-4 |
| +Att-dropout | 27.8 | 89.7 | 83.4 | 79.2 | 87.0 | 77.6 | 77.0 | 86.4 | 61.8 | 62.3 | 73.2 | 4.3e-2 |
| +DropHead | 31.7 | 89.6 | 83.2 | 80.3 | 87.2 | 77.7 | 77.2 | 87.3 | 62.5 | 63.0 | 74.1 | 1.1e-3 |
| +TLM | **35.3** | **90.8** | **83.5** | **81.0** | **87.8** | **78.4** | **77.8** | **87.5** | **63.4** | **64.4** | **75.0** | 2.8e-3 |
| **BERT-base** | | | | | | | | | | | | |
| w/o Att-dropout | 51.0 | 92.3 | **88.2** | 84.2 | 87.7 | 83.5 | 83.2 | 90.3 | 63.0 | 63.0 | 78.6 | 3.1e-3 |
| +Att-dropout | 51.9 | 92.8 | 87.3 | 84.4 | 88.0 | 84.0 | 83.4 | 90.4 | 62.4 | 63.0 | 78.8 | 1.1e-3 |
| +DropHead | 52.0 | **93.4** | 87.8 | **84.5** | 87.5 | 83.6 | 83.1 | 90.4 | 65.2 | 64.4 | 79.2 | 1.8e-3 |
| +TLM | **53.7** | 93.3 | 87.9 | **84.5** | **88.6** | **84.3** | **83.6** | **90.5** | **67.5** | **65.1** | **79.9** | 6.8e-3 |
| **BERT-large** | | | | | | | | | | | | |
| w/o Att-dropout | 59.7 | 93.9 | 88.0 | 86.1 | 88.7 | 86.5 | 85.6 | 92.5 | 69.7 | 63.7 | 81.4 | 2.6e-3 |
| +Att-dropout | 59.8 | **94.3** | 87.9 | 86.5 | 88.9 | 86.6 | 85.7 | 92.7 | 69.6 | 63.7 | 81.6 | 7.9e-4 |
| +DropHead | 60.1 | 94.1 | 88.1 | 85.9 | 89.2 | **86.7** | 85.8 | 92.6 | 70.1 | 64.4 | 81.7 | 6.5e-3 |
| +TLM | **61.0** | 94.2 | **88.6** | **86.5** | **89.3** | **86.7** | **86.1** | **92.8** | **70.8** | **66.4** | **82.2** | 4.7e-4 |

Table 2: Fine-tuned BERT-small, BERT-base, and BERT-large performances on English natural language understanding benchmark GLUE. Each method is tuning with 3 different random seeds. The AVG denotes the average results and STD is the standard deviation of 3 results. The highest numbers are in bold.

| Model | AFQMC | TNEWS1.1 | IFLYTEK | CMNLI | CLUEWSC | CSL | AVG | STD |
|---|---|---|---|---|---|---|---|---|
| BERT -w/o Att-dropout | 73.6 | 56.7 | 60.2 | 79.4 | 62.2 | 80.2 | 68.6 | 2.2e-2 |
| BERT+Att-dropout | 73.7 | 56.6 | 60.3 | **79.7** | 62.1 | 80.4 | 68.8 | 1.1e-3 |
| BERT+DropHead | 73.6 | 57.0 | 60.6 | 79.0 | 71.4 | 80.5 | 70.4 | 5.2e-4 |
| BERT+TLM | **73.8** | **58.2** | **61.5** | 79.3 | **73.4** | **81.4** | **71.3** | 1.1e-3 |

Table 3: Fine-tuned BERT-base performances on Chinese language understanding benchmark CLUE. The AVG denotes the average results and STD is the standard deviation of 3 results.

using BERT-small, BERT-base, and BERT-large to explore the effect on model size. The experiments include BERT without attention-dropout (Att-dropout), with att-dropout/DropHead/TLM at the rate of $10\%/20\%/5\%$ for each task. We then submit all the files of prediction at 3 different random seeds to the official website GLUE[4] and obtain the scores of the test set. As for the hyperparameters, the learning rate, dropout, and batch size are uniformly set to 2e-5, 0.1, and 32 for all the tasks. All the models are trained and evaluated on 24G Nvidia RTX3090.

**Analysis.** We present the specific results in Table 2. Specifically, we can find that there is only a slight increase in the *AVG* (0.2 points) with Att-dropout at small/base/large sizes, and both TLM and DropHead are well ahead of Att-dropout by a large margin. Compared to DropHead,

our method shows a more significant improvement (0.9/0.7/0.5 points) while scaling model size, which provides evidence that our method is effective in improving the understanding of natural language.

Regarding the sentence-level classification, CoLA, which only contains 8,511 sentences in the training set, our method demonstrates a significant improvement from 27.5 to 35.3, 51.0 to 53.7, and 59.7 to 61.7 as we scale the sizes from small, base to large. This finding is coherent with the design principle of TLM for mitigating overfitting. Thus, our method can achieve much larger performance gains than Att-dropout and DropHead as applied to small-scale supervised data. When scaling the size of datasets such as MNLI-m, which contains 392k/9k sentence pairs on training/test sets, our approach still outperforms the DropHead by both 0.7 points at the sizes of BERT-small and BERT–base.

On similarity and paraphrase tasks, such as

[4] https://gluebenchmark.com/

QQP, our method can upgrade the performance by $0.6/1.1$ points ($87.2 \rightarrow 87.8$ and $87.5 \rightarrow 88.6$) compared to DropHead at the sizes of small and base, and this provides evidence that randomly dropping entire heads benefits less than carefully token masking for the representations of sentence-pairs. When scaling the size to BERT-large, the improvement is not obvious like in BERT-base, we speculate that $89.3$ is approaching $91.1$ (the performance of SOTA) on the GLUE leaderboard, where the enhancement is limited only through regularization methods.

As to WNLI, this inference task requires the model to fully understand the contextual information provided by words or phrases in a sentence. The experimental results indicate that our TLM, by carefully controlling masking, can bring more benefits than DropHead and attention dropout.

### 4.2 Chinese Language Understanding

**Dataset.** CLUE, introduced by (Xu et al., 2020), is a widely used benchmark for Chinese Language Understanding Evaluation. Specifically, *TNEWS1.1* task classifies short news titles into 15 categories. As to the IFLTTEK task, which involves assigning a label from a total of 119 categories to app descriptions. Other tasks include *CLUEWSC2020* (CLUEWSC), which determines co-reference between pronouns and nouns in a sentence. AFQMC aims to judge the semantic similarity of sentence pairs. CSL is a keyword recognition task. CMNLI determines the entailment relationship between sentence pairs.

**Model and Training.** We choose the Chinese BERT-base from huggingface[5] as our backbone and train it with/without Attention dropout (Att-dropout), with TLM and DropHead. We submit the results with 3 different random seeds to CLUE leaderboard[6] and obtain the scores. We evaluate our model with the masking rate of $5\%$ and Drophead at the rate of $10\%$. Other hyper-parameters are the same as the original BERT.

**Analysis.** The overall results with TLM, Drop-Head, Att-dropout, and without Att-dropout are reported in table 3. First, both DropHead and our TLM can notably boost the *AVG* compared to with/without Attention dropout, which verifies the effectiveness of these two regularization methods.

Concerning the classification tasks, our approach can significantly outperform DropHead on short text *TNEWS1.1* by $1.2\%$ and long text *IFLTTEK* by $0.9\%$. The possible explanation of promotions is that Siblings-masking and Self-masking introduce a bottleneck, where the networks can only utilize partial neighbors' attention information. Thus, the networks should work hard to become robust and this scheme benefits more than DropHead. We also notice that adding regularization methods results in performance degradation on CMNLI, and we leave this for future work to investigate the reasons.

### 4.3 Chinese Grammatical Error Correction

**Dataset.** Chinese Grammatical Error Correction (CGEC) is a task that automatically detects and corrects the grammatical errors in the input sentence without revising the meaning of the original sentence as much as possible. For this task, we evaluate our proposed method on the benchmark dataset CGED[7] introduced by (Rao et al., 2020). The set contains 28,031/3,115 utterances for training/validation with the average length of sentences 46.24/45.55. For the test sets, CGED2021 and CGED2020 comprise 2,294 and 1,457 utterances.

**Model and Training.** To demonstrate the effectiveness as possible, we choose the strong pre-trained model Chinese Bart[8] as our backbone and fine-tune it with TLM, DropHead, and Att-dropout at the rate of $10\%$. We also compare results with the top-performing baselines GECToR (Omelianchuk et al., 2020) with BERT, RoBERTa, and ELECTRA. The learning rate, batch size, and epoch are set to 3e-5, 32, and 150, respectively.

| Model | Score |
|---|---|
| GECToR-BERT | 32.8 |
| GECToR-RoBERTa | 33.5 |
| GECToR-ELECTRA | 32.7 |
| MAN (Fan et al., 2021) | 41.3 |
| Bart-base -w/o Att-dropout | 42.0 |
| Bart-base + Att-dropout | 42.3 |
| Bart-base + DropHead | 42.7 |
| Bart-base + TLM | **43.7** |

Table 4: The overall results on CGED2021.

[5] https://huggingface.co/bert-base-chinese
[6] https://www.cluebenchmarks.com/

[7] https://github.com/blcuicall/cged_datasets
[8] https://huggingface.co/fnlp/bart-base-chinese

**Analysis.** The definition of metrics is from the Chinese Learner Text Correction[9], and we present the results in Table 4 obtained by the official scripts. First, we can observe that our TLM outperforms GECToR-BERT/RoBERTa/ELECTRA by 10.9/10.2/11.0 points. In contrast with MAN (A more detailed comparison can be found in Appendix C), our approach leads to an improvement of 2.4 points without requiring extra model parameters like MAN. The scale of the set ($28k$ sentences) is relatively small, while the model size of Bart (over $138M$) is quite large, which may easily cause overfitting. Under this circumstance, our method improves by 1.0 and 1.4 points compared to DropHead and attention-dropout. We speculate that DropHead drops entire heads of attention and may lose some syntactic information. Our token-level masking has the advantage of detecting syntax errors and correcting the errors because the absence of some tokens will strengthen the model's sensitivity to syntactic information.

### 4.4 Data-to-Text Generation

**Dataset.** Data-to-text generation is a significant challenge that aims to automatically yield a description to represent the valuable key information in the structured data. The benchmark dataset is ROTOWIRE introduced by (Wiseman et al., 2017), whose output is the summary of NBA basketball games as the input is corresponding records that represent the performance of their teams and players. The summaries are professionally written and relatively well structured with an average generation length of 337.1 words per example. Following the research[10], the set has been split into training, validation, and test sets consisting of 3,398/727/728 summaries, respectively.

**Model and Training.** In this task, we take the encoder-decoder model T5-small[11] as our backbone and compare it with SOTA models. We train T5 with TLM, DropHead, and Att-dropout at the rate of $10\%$. As to the hyper-parameters, we set beam search as 8, learning rate as 5e-4, and max text length as 512. To penalize repetition, the repetition penalty is set to 2.

**Analysis.** We report the results with BLEU in table 5. The current SOTA model HierarchicalEn-

| Model | BLEU |
| --- | --- |
| ENT (Puduppully et al., 2019) | 16.12 |
| DUV (Gong et al., 2020) | 15.92 |
| HierarchicalEncoder (Li et al., 2021) | 17.96 |
| T5 -w/o Att-dropout | 18.00 |
| T5 + Att-dropout | 17.98 |
| T5 + DropHead | 18.01 |
| T5 + TLM | **18.93** |

Table 5: BLEU results on Rotowire.

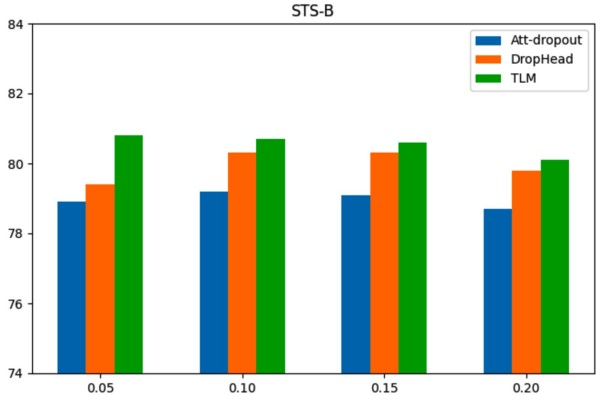

Figure 3: The comparison among Attention Dropout, DropHead, and TLM on STS-B.

coder proposes Number Ranking and Importance Ranking as two auxiliary tasks to capture the individual relations between the records. Our approach increases by 0.97 points relative to HierarchicalEncoder and achieves a new SOTA. Meanwhile, TLM is extremely easy to train with T5 in an end-to-end manner as well as no extra modules or tasks are required. It can also increase by 0.92 points in contrast to DropHead, the advantage for our TLM is the masking scheme can encourage models to capture complex relations among the records and select the salient information in the table. However, dropping entire heads for DropHead may cause the nets are not sensitive in capturing complicated feature relationships.

### 4.5 Ablation study

Although the experimental results are superior, the effectiveness of our TLM has not been thoroughly investigated. Thus, we conduct further studies to gain a better understanding of our approach. As to the selection of datasets, the STS-B and CoLA are sentence-level datasets while IFLYTEK and CSL are long-text datasets, and CGED2020/2021 are grammatical error correction datasets. We hope the tasks can cover different lengths of text meanwhile

---

[9]https://github.com/blcuicall/CCL2022-CLTC/tree/main/metrics/track2

[10]https://github.com/harvardnlp/boxscore-data

[11]https://huggingface.co/t5-small

the diversity can be guaranteed.

**TLM vs Attention Dropout/DropHead.** To analyze the relationship among TLM, attention dropout (Att-dropout), and DropHead applied to self-attention layers in Transformers, we first train BERT-small only with TLM/Att-dropout/DropHead in the [0.05, 0.10, 0.15, 0.20] range to explore their influences on performance. The results are presented in Fig.3. We keep all the parameters the same except for the rates. The finding is that both TLM and DropHead are well ahead of Att-dropout by a large margin on STS-B, and our method is more stable than Drop-Head.

| Method | STS-B |
|---|---|
| BERT w/o regularization | 78.7 |
| + dropout | 78.9 |
| + dropout + Att-dropout | 79.2 |
| + dropout + DropHead | 80.3 |
| + dropout + TLM | **81.0** |
| + dropout + DropHead + Att-dropout | 80.4 |
| + dropout + DropHead + TLM | 80.0 |
| + dropout + TLM + Att-dropout | 80.7 |
| + All | 79.9 |

Table 6: The effect of different regularization combinations. All the rates equal $0.1$.

Second, we test the effect of different combinations on the self-attention models. As shown in Table 6, we observe that adding any type of regularization method can improve the performance of the vanilla model, and our TLM outperforms attention dropout and DropHead under the same conditions by a large margin. When combined together, we find that the performance is not optimal, especially when all regularization methods are used together. This is mainly due to the fact that excessive use of regularization may cause training instability.

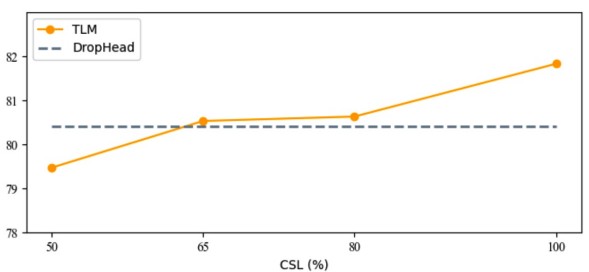

Figure 4: Results with different training data ratios on the test set of CSL.

**Effect on training data ratio.** We further investigated the impact of different training data ratios by training BERT-base+TLM using 50%/65%/80% of supervised data on CSL, and the results are presented in Fig.4. In contrast to DropHead, TLM can achieve comparable performances even when trained with only 50% of data. Moreover, our method outperforms DropHead with 65% of data on CSL. The improvements may be attributed to the token-level masking, as this strategy encourages the model to capture meaningful context representation with long input sentences. Therefore, our TLM can benefit the robustness of the networks and reduce the dependence on supervised data.

| Model | CGED2020 | | | | |
|---|---|---|---|---|---|
| | $E_R$ | $D_R$ | DET-F1 | COR-F1 | SCORE |
| Bart | - | - | 80.5 | 19.3 | 37.8 |
| TLM | 20 | 20 | 78.9 | 19.3 | 37.1 |
| TLM | 15 | 15 | 81.4 | 19.5 | 38.3 |
| TLM | 15 | 10 | 82.1 | 20.1 | 39.1 |
| TLM | 10 | 15 | **82.4** | 20.2 | **39.2** |
| TLM | 10 | 10 | 82.1 | **20.3** | 38.9 |

Table 7: The results on CGEC test sets of CGED2020. The $E_R$ and $D_R$ refer to the rate of masking in the encoder and decoder. DET-F1 means the F1 score for detection, and COR-F1 denotes the F1 score for correction. The complete results can be found in Appendix B.

**Effect of TLM rate.** We conduct further testing on the impact of varying the rate of masking in the encoder and decoder of Bart-base, ranging from 10% to 20%. As outlined in Table 7, the results of *SCORE* are better than the baseline, except for the rate of 20%. A possible explanation for why our TLM underperforms BERT at the rate of 20% is that an excessive amount of masking may confuse the networks and decrease their ability to comprehend syntax information. It's important to note that a too-large rate should be cautious. Overall, our optimal option is the group of (10%, 15%) for encoder and decoder, and it outperforms the strong baseline by 1.4 points ($37.8 \rightarrow 39.2$). This promotion demonstrates that our TLM can enhance the understanding of grammatical information and guide the model toward correcting grammatical errors.

**Effect of Siblings-masking/Self-masking.** We also analyze the influence of our masking techniques, and the results at the size of BERT-small are reported in Table 8. It can be observed that the

results decrease by 2.6% and 0.6% on CoLA and STS-B when removing the Siblings-masking technique. Similarly, the results without Self-masking decrease by 3.0% and 1.2%, and there is a drop of 7.6% and 2.2% without both techniques. These findings highlight the importance of both Siblings-masking and Self-masking methods.

| Model | CoLA | STS-B |
|---|---|---|
| TLM | **35.3** | **81.0** |
|     -w/o Siblings-masking | 32.7 | 80.5 |
|     -w/o Self-masking | 32.3 | 79.9 |
|     -w/o Both | 27.7 | 78.9 |

Table 8: The effect of our masking techniques.

## Conclusion

In this paper, we propose a simple training strategy, Token-Level Masking (called TLM) to reformulate the computation flow of multi-head self-attention for reducing overfitting. During training, we randomly invoke one of the two masking techniques: 1) Siblings-masking, where the masked token is forbidden to interact with its siblings when computing attention weights, and 2) Self-masking, the attention weights for the masked token is solely reliant on others. This regularization scheme enables the networks to work hard to become robust and acquire meaningful information.

To verify the effectiveness of our proposed method, we conducted various experiments with 18 benchmark datasets. The results demonstrate that TLM can consistently improve the performances compared to the strong baselines and even achieve SOTA on data-to-text generation. Through further analysis, we observe that our TLM is more stable than DropHead or attention dropout. Meanwhile, it can seamlessly integrate with pre-trained models.

## Limitations

Here, we list several of what we consider to be limitations:

1. The rate of our masking is a hyper-parameter that needs to be tuned, as the experiments shown in Table 7, the performance may underperform when the rate is set too large (e.g., over 20%).

2. We argue that TLM can also be applied to vision or speech Transformer-based networks, such as VIT (Dosovitskiy et al., 2020) and UniSpeech (Wang et al., 2021), we leave it as the future work for further validation. Meanwhile, we haven't yet estimated the performance by combining TLM with extremely large language models, such as T5-11B and LLaMA.

3. Due to the limitation of computational resources. we merely fine-tuned the pre-trained models with TLM in this work. The effectiveness of TLM applied to the pre-training phase needs to be further validated.

4. In contrast with dropout, TLM can only apply to Transformer-based networks, not all the neural networks, such as CNN or LSTM.

5. Despite numerous ablation studies being performed, the explanation of TLM's optimization on the self-attention mechanism remains insufficient, especially in terms of the effect on attention distribution. Further exploration is needed.

## Acknowledgements

The authors gratefully acknowledge Yao Zhao, Min Liang, Mengqi Zhang, Xiangyu Jin, Fengli Shi, Shanhoo Luo, and Fang Shu for giving valuable suggestions on this study. Our thanks also go to all the anonymous reviewers for their positive feedback. The work is supported by the National Key Research and Development Project of China (2022YFF0902000). In addition, we thank HANGZHOU YIYOULIAO TECHNOLOGY CO LTD https://www.yiyouliao.com/ for providing computing resources.

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

# A Appendix

**Task descriptions on GLUE.** We provide more details task descriptions and statistics in Fig.5.

| Model | CGED2021 | | |
|-------|----------|----|-----|
| | Detection-F1 | Correction-F1 | SCORE |
| MAN | 77.6 | **30.9** | 41.3 |
| TLM | **77.8** | 30.7 | **43.7** |

| Model | CGED2020 | | |
|-------|----------|----|-----|
| | Detection-F1 | Correction-F1 | SCORE |
| MAN | 80.5 | 19.8 | 37.2 |
| TLM | **82.1** | **20.3** | **38.9** |

Table 9: The comparisons between MAN and TLM (10%) on CGEC test sets of CGED2021 and CGED2021.

**Experimental settings on CLUE.** As described in Section 4.2, we change the max epochs and mask rate during training for better scores, and the settings are listed in Table 10.

# B Appendix

**The detailed results on Chinese Grammatical Error Correction.** All the detailed results are listed in Table 11 compared with top-performers. In contrast to encoder-based Transformer models e.g., GECToR-BERT/RoBERTa/ELECTRA, our proposed method can outperform them by a large margin in all the metrics. For example, the masking rate of (10%, 10%) has much higher Detection, and Correction F1 scores, which demonstrates TLM has the advantage of accurately locating grammatical errors and successfully correcting these errors. For the strong baseline Bart-base, the large improvement lies in Correction F1 (from 28.2 to 30.7).

# C Appendix

**In contrast with MAN.** Mask Attention Networks (MAN) introduces a new layer to capture the relationship between the Self-Attention Network and Feed-Forward Network, causing the model parameters to substantially increase and hindering its wide application with pre-trained language models. TLM doesn't have these issues, which simply modifies the calculation workflow of self-attention to enhance the contextual information without extra modules. We run the code from `https://github.com/LibertFan/MAN/tree/main/summarization` and make no modifications, except for using sentencepiece[12] as the tokenizer for Grammatical Error Correction

---

[12] `https://github.com/google/sentencepiece`

| Corpus | \|Train\| | \|Test\| | Task | Metrics | Domain |
|--------|-----------|----------|------|---------|--------|
| | | | Single-Sentence Tasks | | |
| CoLA | 8.5k | **1k** | acceptability | Matthews corr. | misc. |
| SST-2 | 67k | 1.8k | sentiment | acc. | movie reviews |
| | | | Similarity and Paraphrase Tasks | | |
| MRPC | 3.7k | 1.7k | paraphrase | acc./F1 | news |
| STS-B | 7k | 1.4k | sentence similarity | Pearson/Spearman corr. | misc. |
| QQP | 364k | **391k** | paraphrase | acc./F1 | social QA questions |
| | | | Inference Tasks | | |
| MNLI | 393k | **20k** | NLI | matched acc./mismatched acc. | misc. |
| QNLI | 105k | 5.4k | QA/NLI | acc. | Wikipedia |
| RTE | 2.5k | 3k | NLI | acc. | news, Wikipedia |
| WNLI | 634 | **146** | coreference/NLI | acc. | fiction books |

Figure 5: Task descriptions and statistics of GLUE.

| Parameters | AFQMC | TNEWS | IFLYTEK | CMNLI | WSC | CSL |
|------------|-------|-------|---------|-------|-----|-----|
| Learning Rate | 2e-5 | 2e-5 | 2e-5 | 3e-5 | 1e-5 | 1e-5 |
| Max Epoch | 9 | 6 | 9 | 9 | 50 | 9 |
| Batch Size | 16 | 16 | 16 | 16 | 16 | 8 |
| Dropout | 0.1 | 0.1 | 0.1 | 0.1 | 0.1 | 0.1 |
| Mask Rate | 0.2 | 0.1 | 0.15 | 0.1 | 0.1 | 0.1 |

Table 10: The settings of our TLM on Chinese Language Understanding benchmark CLUE.

task, and the results are listed in Table 9. Clearly, TLM can substantially increase by 1.7 points compared to MAN on CGED2020 (37.2 → 38.9) and 2.4 points on CGED2021 (41.3 → 43.7), which verifies the design of our TLM that masking some connections in the attention calculation can steer the networks to become powerful.

# D   Appendix

**The proportion of Siblings-masking/Self-masking.** To further investigate the contributions of different proportions of Siblings-masking and Self-masking, we perform experiments in the group with [0/100%, 25%/75%, 50%/50%, 75%/25%, 100%/0] at rates range in [5%, 10%, 15%] for Siblings-masking and Self-masking on RTE. The results are reported in Table 12, we note that the experiments with only RTE are inadequate, thus we would perform further experiments and report the results on our GitHub https://github.com/Young1993/tlm. As shown in table 12, 50-50 chance is the relatively better choice compared to 0/100% or 100%/0, and we recommend using this default setting for simplicity.

| Model | CGED2021 | | | | | | | |
|---|---|---|---|---|---|---|---|---|
| | ENC-R | DEC-R | FPR | Detection | Identification | Position | Correction | Score |
| GECToR-BERT | - | - | 31.9 | 74.5 | 46.3 | 27.5 | 14.6 | 32.8 |
| GECToR-RoBERTa | - | - | 30.2 | 74.3 | 46.8 | 27.8 | 15.3 | 33.5 |
| GECToR-ELECTRA | - | - | 29.5 | 73.1 | 45.7 | 27.6 | 14.0 | 32.7 |
| Bart-base | - | - | 22.3 | 77.4 | 54.3 | 31.8 | 28.2 | 42.3 |
| Bart-base + DropHead | 10 | 10 | 22.7 | 76.7 | 54.5 | 33.0 | 29.2 | 42.7 |
| Bart-base + TLM | 20 | 20 | **21.3** | 76.0 | 53.6 | 30.9 | 28.1 | 41.8 |
| Bart-base + TLM | 15 | 15 | 22.7 | 77.3 | 55.7 | 33.4 | 29.9 | 43.4 |
| Bart-base + TLM | 15 | 10 | 23.6 | 77.7 | **55.9** | 34.0 | 30.4 | 43.6 |
| Bart-base + TLM | 10 | 15 | 22.9 | 77.7 | 55.4 | **33.6** | 30.0 | 43.4 |
| Bart-base + TLM | 10 | 10 | 23.4 | **77.8** | **55.9** | 30.7 | **30.7** | **43.7** |

Table 11: The detailed results on Chinese Grammatical Error Correction benchmark test set of CGED2021. The ENC-R and EDC-R refer to the rate of masking in the encoder (ENC) and decoder (DEC),e.g., 20 denotes masking 20% tokens. FPR, Detection, Identification, Position, and Correction denote false positive rate, detection-F1, identification-F1, position-F1, and correction-F1, respectively. We collected the results of GECToR-BERT/RoBERTa/ELECTRA from `https://github.com/blcuicall/CCL2022-CLTC/tree/main/baselines/track2`.

| Rate | 0/100% | 25%/75% | 50%/50% | 75%/25% | 100%/0 |
|---|---|---|---|---|---|
| 5% | 65.6 | 66.2 | 67.5 | 66.1 | 66.9 |
| 10% | 66.9 | 66.5 | 67.1 | 66.3 | 65.3 |
| 15% | 64.1 | 63.9 | 64.3 | 64.2 | 63.8 |

Table 12: Different proportions of Siblings-masking and Self-masking on RTE.