# OpenReview forum: "TLM: Token-Level Masking for Transformers"
_EMNLP/2023/Conference — EMNLP 2023 Main_

### Official Review · Reviewer_NQeB · 2023-08-03

**Soundness:** 3

**Excitement:**

3: Ambivalent: It has merits (e.g., it reports state-of-the-art results, the idea is nice), but there are key weaknesses (e.g., it describes incremental work), and it can significantly benefit from another round of revision. However, I won't object to accepting it if my co-reviewers champion it.

**Paper Topic And Main Contributions:**

This paper introduces a novel regularization technique called "Token-Level Masking (TLM)." This is a specially controlled version of attention dropout, where attention weights to particular tokens from other context tokens (sibling masking) or themselves (self-masking) are dropped during training.
Experiments demonstrate that this technique leads to better performance across several tasks, such as GLUE.
Further analysis tries to know the characteristics of the TLM concerning the simultaneous use of TLM with other regularization techniques and hyperparameters (Tables 6 and 7) and elucidate the effect of each component in the proposed method (Table 8)

**Questions For The Authors:**

I hope the authors answer the questions mentioned in Reasons To Reject.
Why specific baseline is discarded in specific experiments (e.g., BERT+Att-dropout in Tables 3--5)?

**Reasons To Accept:**

- The idea is simple, and integrating TLM into Transformer variants is easy to implement.
- Experiments have a broad range of tasks, and the improvement by TLM seems consistent across these experiments.
- The ablation study and analysis (Section 4.5) cover the points the reader may care about; I would like to know the generality of the observations, though.

**Reasons To Reject:**

- Although the coverage of tasks in the experiments seems sufficient, the variation of the tested model variants is limited. Specifically, only a different single model (e.g., BERT-base) is examined in different experiments.
- How many models with different seeds are used in the experiments? If each experiment relies on a single run, I suspect the generality of the improvement by TLM and the significance of the performance differences.
- Ablation study and some analyses (Section 4.5) are also conducted only in specific tasks (e.g., STS-B) or input. I understand that conducting all the analyses in all the tasks requires a high computation cost, but I would like to see a motivation for why the specific task/model was used in each analysis, and at least the information in this paper alone seems to cherry-pick the good results. Furthermore, I am not sure what generalizable findings can be obtained by seeing the attention pattern in a single instance (Figure 4).

To sum up, the focus of the experiments seemed to be blurred (in terms of model and task types), with sporadic evidence of effectiveness for certain combinations of tasks and models. In other words, I'd like to know the motivation why each setting was chosen. I am willing to accept reduced task coverage, so I would like to see more solid findings with respect to, especially, model size/variants/seeds and the characteristics/limitations of the proposed method (the robustness of the results in Tables 6--8 are).

**Reproducibility:**

3: Could reproduce the results with some difficulty. The settings of parameters are underspecified or subjectively determined; the training/evaluation data are not widely available.

**Reviewer Confidence:**

3: Pretty sure, but there's a chance I missed something. Although I have a good feel for this area in general, I did not carefully check the paper's details, e.g., the math, experimental design, or novelty.

**Typos Grammar Style And Presentation Improvements:**

"Score" in Tables 2--4 seems too abstract a description. Those in Tables 2 and 3 seem to be a macro average of the performance across the component tasks; this should be explained.
The text on the graphs is small. In particular, the y-axis is somewhat misleading because it does not start from 0.

---

> ### Author Rebuttal · Authors · 2023-08-28
>
> Dear reviewer, we appreciate all your detailed comments.
>
> For the question in “Reasons To Reject:”:
> - As to the question “the variation of the tested model variants is limited…”. In our work, we have tested “BERT, BART, and T5”, including small- and base-size. Saying that “The variation of the tested model is limited” is a little strict, and we beg the reviewer could check the details again.
> - For GLUE and CLUE, we just trained the pre-trained model once and submitted the files of predictions to the official leaderboard website to obtain the performances. We thought the results of such multiple tasks could provide some indication of generalizability. We then conducted 3 times experiments on GLUE with BERT-small/BERT-base and obtained the performances from the official leaderboard website. The results are reported in the official comments (Table 2 in the revised manuscript will be updated), and we hope that the additional experiments can alleviate the reviewer's concerns.
> - For the question of “Ablation study and some analyses (Section 4.5) are also conducted only in specific tasks (e.g., STS-B) or input.”, the reason is that STS-B and CoLA are sentence-level datasets, IFLYTEK and CSL are long text datasets, and CGED2020/2021 are grammatical errors datasets. We just hope the tasks can cover different lengths of text meanwhile the diversity can be guaranteed.
> - For the question of “ …seeing the attention pattern in a single instance (Figure 4).”, we present 2 instances with all layers of attention distribution in Appendix C and D, and we kindly request the reviewer may look again at the Appendix. The finding is that TLM is less affected by certain words because the attention distributions are aware in different layers (see Appendix C).
>
> The focus of our experiments is to verify the generality of our TLM for reducing overfitting. We totally agree that “…willing to accept reduced task coverage, so I would like to see more solid findings with respect to, especially, model size/variants/seeds and the characteristics/limitations”. The comparison of experiments needs to highlight the strengths of our work, as well as the limitations. We will carefully modify these in the revised manuscript.
>
> For the question in “Questions For The Authors”:
> - As to the question ” Why specific baseline is discarded in specific experiments (e.g., BERT+Att-dropout in Tables 3--5)?”, Both our experiments and DropHead indicate that the difference is not obvious with or without attention dropout, this leaves us careless not comparing attention dropout in Tables 3—5, we have reported the results on CLUE for easing the reviewer’s worries.
> |Model | AFQMC | TNEWS1.1 | IFLYTEK  | CMNLI | CLUEWSC | CSL | Score|
> |  ----  | ----  | ---- | ----  | ----  | ---- | ----  | ----  |
> |  BERT-base -w/o Att-dropout    | 73.5  | 56.7 | 60.1 |79.5 | 62.2 |	80.3 | 68.7 |
> |  BERT-base + Att-dropout         | 73.7  |  56.6 | 60.3  |79.7| 62.0 |80.4  | 68.8 |
> |  BERT-base+DropHeadt            | 73.6  | 57.0  |60.6  |78.9 | 71.6 | 80.4  | 70.4|
> |  BERT-base+TLM                      | 73.9   | 58.1 | 61.5 | 79.3 | 73.4 |81.6 | 71.3 |
>
> Table 3: Fine-tuned BERT-base performances on Chinese language understanding benchmark CLUE.
>
> The results show that the BERT-base with attention-dropout underperforms our TLM. Because the schedule is tight, we will do more experimental comparisons in the revised version.
>
> We will carefully revise the Typos and grammar problems in the revised manuscript.

---

### Official Review · Reviewer_2Lox · 2023-08-05

**Soundness:** 3

**Excitement:**

3: Ambivalent: It has merits (e.g., it reports state-of-the-art results, the idea is nice), but there are key weaknesses (e.g., it describes incremental work), and it can significantly benefit from another round of revision. However, I won't object to accepting it if my co-reviewers champion it.

**Paper Topic And Main Contributions:**

This paper introduces token level masking (TLM), a regularization approach for training Transformer models. TLM randomly mask tokens in the attention layer and force the model to exploit partial neighbors' information when producing representation. The authors conduct extensive evaluation on both English (GLUE) and some Chinese NLU benchmarks and show some improvements upon conventional attention dropout as well as advanced methods such as Drophead.

**Questions For The Authors:**

Why BERT-small is used for GLUE tasks?

**Reasons To Accept:**

1. The paper is overall well-written and easy to follow.
2. The idea of masking tokens in attention layers is novel, simple, and intuitive.
3. The authors conduct extensive experiments on many datasets including both English and Chinese datasets and many different models including BERT, BART, T5, etc., and show consistent improvements.
4. The ablation studies and analyses are well conducted and provide some insights for Transformer training regularization.

**Reasons To Reject:**

1. The authors do not conduct a hyperparameter search or do not mention the search detail. Considering that different regularization methods may result in substantial changes on optimal values for other hyperparameters, it is unclear if the experiments are solid enough.
2. The most representative experiments should be on the GLUE benchmark, however, only BERT-small results are presented, which is very confuse because standard practice all conduct experiments on GLUE with at least BERT-base.

**Reproducibility:**

4: Could mostly reproduce the results, but there may be some variation because of sample variance or minor variations in their interpretation of the protocol or method.

**Reviewer Confidence:**

4: Quite sure. I tried to check the important points carefully. It's unlikely, though conceivable, that I missed something that should affect my ratings.

---

> ### Author Rebuttal · Authors · 2023-08-28
>
> Dear reviewer, thanks for your constructive comments.
>
> - As to question 1.” The authors do not conduct a hyperparameter search…”, it’s our negligence without a clear explanation for hyperparameters, which is the rates of attention-dropout/DropHead/TLM in line 266. Here, we choose the best choice of rates for different regularization methods (The choices followed DropHead \cite{zhou-etal-2020-scheduled} for baselines). We also conducted detailed experiments to compare the hyperparameters between attention-dropout /DropHead/TLM in the Ablation study (Line 431), and the results indicated our method is more powerful than the baselines with different hyperparameters.
>
> - For the question 2. “Why BERT-small is used for GLUE tasks?” I’m so sorry for our negligence of model size. Because the schedule is overly tight, we have reported the results of BERT-base in the official comments, and we will report the results of BERT-large in the revised manuscript.
>
> We will carefully revise these issues in the revised manuscript.

---

### Official Review · Reviewer_Lwg8 · 2023-08-07

**Soundness:** 4

**Excitement:**

4: Strong: This paper deepens the understanding of some phenomenon or lowers the barriers to an existing research direction.

**Paper Topic And Main Contributions:**

In Transformer models, structured dropout approaches such as attention dropout and DropHead have been used to regularize multi-head attention mechanisms. This paper proposes a novel regularization, token-level masking (TLM), based on the token level instead of the structural level. Its idea is to carefully mask partial connections between tokens in the attention calculation, rather than dropping whole attention heads or random attention weights.

The effectiveness of TLM has been thoroughly evaluated through experiments using a wide range of models on four different NLP tasks across 18 datasets. TLM is simple and easy to implement yet outperforms existing baseline methods in most cases. New SOTA was achieved for a task. Interesting ablation studies was provided.

**Questions For The Authors:**

- A. In ll.88-90 you state "In this manner, the networks are forced to utilize partial neighbors' attention information, not the whole. Can you explain this logic in more detail?
- B. Can the performance improvements in Section 4.1 (English Language Understanding) be seen not only in BERT-small, but also in BERT-base and BERT-large? And how big is the improvement margins?
- C. What exactly is the "Scale" after the inner product of query and key in the formula at l.174?
- D. Does it make any sense to choose a token to mask for each layer and have the same mask token for each head? What do you think the performance and robustness are compared to masking different tokens for each head?
- E. If TLM was used in experiments with encoder-decoder models such as T5, how was cross attention regularized?

**Reasons To Accept:**

- Simple and easy to implement attention regularization approach
- Thorough experiments evaluating on a wide range of data, tasks, and models
- Proposed method shows performance improvement over existing methods in most settings
- Sufficient and informative ablation studies
- Achieved new SOTA on data-to-text generation task (Rotowire)
- Clearly stated connections to existing methods
- Well written and easy to understand

**Reasons To Reject:**

- I did not understand the logic of one of the methodology explanations (Question A).
- Section 3.2 (explanation of the approach) has a lack of information, and important information about the method is scattered in other sections.
    - Only self-masking is explained and there is no explanation of siblings-masking.
    - There is no description of selecting one of the two masking strategies for each batch in the 50-50 chance.
- Model choice for the evaluation on English language understanding (Section 4.1) is somewhat unnatural (Question B).

**Reproducibility:**

4: Could mostly reproduce the results, but there may be some variation because of sample variance or minor variations in their interpretation of the protocol or method.

**Reviewer Confidence:**

5: Positive that my evaluation is correct. I read the paper very carefully and I am very familiar with related work.

**Typos Grammar Style And Presentation Improvements:**

- ll.50-54: Grammatically broken
- Last line of Table 1 "w/o Attention" -> "w/o Attention-dropout"
- l.73: no need to insert space between the werd "function" and footnote number "1"
- Section name of Section 3.1: "Transformer" -> "(Multi-head) Attention in Transformer"
- It would be easier to explain if the shapes of query Q, key K, and value V are explicitly indicated.

---

> ### Author Rebuttal · Authors · 2023-08-28
>
> Dear reviewer, we are very grateful for your positive comments.
>
> - For the question A. We deeply apologize for the unclear statement. Because of the existence of Siblings-masking and Self-masking, the connections between the masked tokens and their neighboring tokens are invalid, which are implemented by assigning a large negative number in the matrix of attention weights.
> - As to the question B. I’m so sorry for our gross negligence that the experiments of GLUE without base- and large-size BERT, we can only report the results on GLUE with the base-size in the official comments due to the overly tight schedule.
> - For the question C. We are deeply apologized for the mistake of the "Scale", which is {\sqrt{d_{emb} / H}}, we should remove “Scale” in Eq.2 because {\sqrt{d_{emb} / H}} is computed in Eq.2.
> - For the question D. We think it does make sense and we can conduct the experiments to compare the performances.
> - For the question E. The masking of cross-attention is like self-attention, the difference is the query comes from the decoder layer, and the memory keys/values come from the output of the encoder. We will test it and report the results in the revised manuscript.
>
> We will carefully revise the Typos and grammar problems in the revised manuscript.

---

### Official Review · Reviewer_e4DL · 2023-08-12

**Soundness:** 4

**Excitement:**

4: Strong: This paper deepens the understanding of some phenomenon or lowers the barriers to an existing research direction.

**Paper Topic And Main Contributions:**

The paper proposes a token-level masking (TLM) scheme as a regularization method at the training stage to mitigate the overfitting problem. TLM selects random tokens for each layer in the transformer. For the selected tokens, it has two choices for computing attention score: 1) sibling masking, where neighboring tokens contribute to attention scores independently of the selected tokens, and 2) self-masking, where attention scores are calculated without accounting for their own impacts. It shows that training with TLM can improve the performance of various downstream tasks compared to vanilla training.

Contributions:  a new regularization technique, NLP engineering experiments

**Questions For The Authors:**

1. Does the token-level masking happens at fine-tuning or pre-training for all experiments? I assume TLM happens at the fine-tuning stage for all experiments. It would be better to explicitly say that in the experimental section. How about applying TLM at the pre-training stage?
2. Since some of the improvements are marginal, how many trials have the authors run for each experiment? Including mean and standard deviation in the paper would be helpful.
3. Why specifically have Siblings-masking and Self-masking at 50-50 chance?

**Reasons To Accept:**

1. Simple attention regularization methods which can be easily integrated into the training of any LLMs.
2. Comprehensive experiments and ablation study across different model architectures (encoder-only, encoder-decoder) and different datasets (English/Chinese) to show the TLM improves the downstream task performance.
3. This regularization method enhances training efficiency since it requires less amount of training data as analyzed in section 4.5.

**Reasons To Reject:**

1. The motivation and novelty are unclear to me and a more detailed explanation is needed. Are there any intuitions of this token-level masking improves performance? How much difference compared to masking at the input level? Is it designed for fine-tuning or pre-training?
2. The improvements are mostly shown for small or base-size models given the experiments in the paper. It is not clear whether TLM is still useful for larger or more powerful models.

**Reproducibility:**

4: Could mostly reproduce the results, but there may be some variation because of sample variance or minor variations in their interpretation of the protocol or method.

**Reviewer Confidence:**

3: Pretty sure, but there's a chance I missed something. Although I have a good feel for this area in general, I did not carefully check the paper's details, e.g., the math, experimental design, or novelty.

---

> ### Author Rebuttal · Authors · 2023-08-26
>
> Dear reviewer, we deeply appreciate all your constructive comments and advice.
>
> For the question in “Reasons To Reject:”:
> - ” The motivation and novelty”, our motivation is introducing a new regularization scheme to manipulate the connections between tokens to further mitigate overfitting for Transformers. To the best of our knowledge, the idea of our regularization by masking tokens in multi-head attention is novel.
> - I guess the question of “How much difference compared to masking at the input level?” is to act on the embedding layer, our method is only applied to the attention layer thus it cannot be utilized in the embedding layer.
> - Our approach can be used both in fine-tuning and pre-training. We only estimated the models in the fine-tuning phase due to the limitation of computational resources (We declared this in the Limitation). We are so sorry for the unclear and inapparent declaration.
> - "The improvements are mostly shown for small or base- …...” is our gross negligence, and we will report the results of the large model in the revised manuscript.
>
> For the question in “Questions For The Authors:”:
> - 1.	We use TLM at the fine-tuning stage for all experiments, and it’s our overlook to only state in the Limitation (Line 543). We would test it at the pre-training stage for further validation.
> - 2.	For GLUE and CLUE, we only run once and obtain the results from the official leaderboard website. Other experiments are running 2 times and the best of them are reported. To ease your concern, we reported the mean results and standard deviation on GLUE with small- and base- sizes in the official comments. The revised manuscript would cover the large size due to the overly tight schedule.
> - 3.	For the rate of Siblings-masking and Self-masking, it’s our overlook to simply use the group of 50-50 chance, we would test other rates for ablations and report the results in the revised manuscript.

---

### Meta-Review · Area_Chair_WWfX · 2023-09-18

**Recommendation:** 5

**Metareview:**

The paper introduces Token-Level Masking (TLM), a novel regularization technique for training Transformer models. TLM involves masking tokens in the attention layer, forcing models to exploit partial neighbor information for improved representation. Extensive experiments on various NLP tasks demonstrate the effectiveness of TLM compared to existing methods.

Reviewers had several concerns, including the need for a more detailed explanation of TLM's motivation and its applicability to larger models. They questioned the lack of hyperparameter search details, the choice of specific models for experiments, and the variation in tested model variants. Additionally, they sought clarification on the selection of specific tasks and models for ablation studies. The authors clarified TLM's motivation, addressed the issue of hyperparameters, and committed to providing results for larger models. They explained their choices for specific tasks and models in ablation studies and expressed willingness to conduct more experiments to address baseline comparisons. The authors have demonstrated a proactive approach to improving the paper in the revised version.

---

### Decision · Program_Chairs · 2023-10-07

**Decision:**

Accept-Main

**Comment:**

The paper introduces Token-Level Masking (TLM), a novel regularization technique for training Transformer models. TLM involves masking tokens in the attention layer, forcing models to exploit partial neighbor information for improved representation. Extensive experiments on various NLP tasks demonstrate the effectiveness of TLM compared to existing methods.

Reviewers had several concerns, including the need for a more detailed explanation of TLM's motivation and its applicability to larger models. They questioned the lack of hyperparameter search details, the choice of specific models for experiments, and the variation in tested model variants. Additionally, they sought clarification on the selection of specific tasks and models for ablation studies. The authors clarified TLM's motivation, addressed the issue of hyperparameters, and committed to providing results for larger models. They explained their choices for specific tasks and models in ablation studies and expressed willingness to conduct more experiments to address baseline comparisons. The authors have demonstrated a proactive approach to improving the paper in the revised version.